# Comparative Analysis of Five Multiplex RT-PCR Assays in the Screening of SARS-CoV-2 Variants

**DOI:** 10.3390/microorganisms10020306

**Published:** 2022-01-27

**Authors:** Vanessa De Pace, Bianca Bruzzone, Andrea Orsi, Valentina Ricucci, Alexander Domnich, Giulia Guarona, Nadia Randazzo, Federica Stefanelli, Enrico Battolla, Pier Andrea Dusi, Flavia Lillo, Giancarlo Icardi

**Affiliations:** 1Hygiene Unit, Ospedale Policlinico San Martino—IRCCS, 16132 Genoa, Italy; bianca.bruzzone@hsanmartino.it (B.B.); valentina.ricucci@hsanmartino.it (V.R.); alexander.domnich@hsanmartino.it (A.D.); nadia.randazzo@hsanmartino.it (N.R.); federica.stefanelli@hsanmartino.it (F.S.); 2Department of Health Sciences (DISSAL), University of Genoa, 16132 Genoa, Italy; andrea.orsi@unige.it (A.O.); giuly.guarons@outlook.it (G.G.); icardi@unige.it (G.I.); 3Division of Clinical Pathology, Azienda Sanitaria Locale n°5, 19121 La Spezia, Italy; enrico.battolla@asl5.liguria.it; 4Microbiology Department, Sanremo Hospital, 18038 Imperia, Italy; a.dusi@asl1.liguria.it; 5Laboratory of Clinical Pathology, ASL2 Savonese, 17100 Savona, Italy; f.lillo@asl2.liguria.it

**Keywords:** diagnostics, pandemic, surveillance

## Abstract

The rapid and presumptive detection of SARS-CoV-2 variants may be performed using multiplex RT-PCR assays. The aim of this study was to evaluate the diagnostic performance of five qualitative RT-PCR tests as compared with next-generation sequencing (NGS). We retrospectively examined a multi-variant panel (*n* = 72) of SARS-CoV-2-positive nasopharyngeal swabs categorized as variants of concern (Alpha, Beta, Gamma and Delta), variants under monitoring (Iota and Kappa) and wild-type strains circulating in Liguria (Italy) from January to August 2021. First, NGS libraries of study samples were prepared and mapped to the reference genome. Then, specimens were screened for the detection of L452R, W152C, K417T, K417N, E484Q, E484K and N501Y mutations using the SARS-CoV-2 Variants II Assay Allplex, UltraGene Assay SARS-CoV-2 452R & 484K & 484Q Mutations V1, COVID-19 Ultra Variant Catcher, SARS-CoV-2 Extended ELITe MGB and Simplexa SARS-CoV-2 Variants Direct. The overall accuracy of these assays ranged from 96.9% to 100%. Specificity and sensitivity were 100% and 96–100%, respectively. We highly recommend the use of these assays as second-level tests in the routine workflow of SARS-CoV-2 laboratory diagnostics, as they are accurate, user friendly, low cost, may identify specific mutations in about 2–3 h and, therefore, optimize the surveillance of SARS-CoV-2 variants.

## 1. Introduction

Since the onset of the coronavirus disease 2019 (COVID-19) pandemic, multiple emerging variants of severe acute respiratory syndrome coronavirus 2 (SARS-CoV-2) have been identified, but only some of these have a high degree of global public health significance [1,2]. The scientific research community has categorized these variants as variants of concern (VOCs), variants under monitoring (VUMs) and variants of interest (VOIs) [1,2].

VOCs are associated with an increased transmission or virulence, high levels of hospitalization or deaths as a result of severe disease, reduced antibody neutralization for previously infected or vaccinated individuals, decreased efficacy of therapeutics or vaccines and a greater ability to escape from diagnostic detection [3,4,5,6,7,8]. VOIs present genetic changes that could affect some virus features, such as transmissibility; disease severity; and immune, diagnostic and therapeutic escapes. The increase in virus transmission observed in multiple countries suggests that these variants could be an emerging risk to global public health [2]. Finally, VUMs are variants with important substitutions that affect viral characteristics without any evidence of phenotypic or epidemiological impact. These variants, including those previously designated as VOCs or VOIs, present a minor threat to public health [2].

Based on recent epidemiological data (as of November 2021), according to the World Health Organization (WHO), the following SARS-CoV-2 variants meet the definition of VOCs: Alpha (B.1.1.7), Beta (B.1.351), Gamma (P.1), Delta (B.1.617.2) and Omicron (B.1.1.529). VOIs are represented by Lambda (C.37) and Mu (B.1.621), while VUMs are Eta (B.1.525), Iota (B.1.526) and Kappa (B.1.617.1) variants. The lineages AZ.5, C.1.2, B.1.630 and B.1.640 lack a WHO label [2].

These variants have increased the diagnostic activity for SARS-CoV-2, which is committed to diagnosing and controlling infection, including activities on the viral genome sequencing for epidemiological surveillance, COVID-19 patients with reinfections or vaccination failures and outbreak investigations.

In order to rapidly detect these variants, the diagnostic performances of five different real-time reverse-transcription polymerase chain reaction (RT-PCR) assays were evaluated, and their results were compared with next-generation sequencing (NGS).

## 2. Materials and Methods

We retrospectively examined a panel of VOCs and VUMs selected from monthly collections of the “Quick Survey on SARS-CoV-2 Variants” supported by the Italian Institute of Health and the Ministry of Health. These specimens were collected between January and August 2021. In this study, a total of 72 SARS-CoV-2-positive samples were analyzed by the Hygiene Unit of San Martino Hospital (Genoa), which is the regional reference laboratory, the Microbiology Department of Sanremo Hospital (Imperia) and the Division of Clinical Pathology of the Local Health Unit 5 (La Spezia). Each specimen was collected and immediately tested for SARS-CoV-2 diagnosis. Then, the total RNA was automatically extracted and processed for NGS; full genome was obtained for 69 cases, while the S gene was sequenced for the remaining 3 cases. NGS required approximately 2–3 days. All molecular screening tests for SARS-CoV-2 variants were performed following the sequencing. The multi-variant panel was defined by selecting different SARS-CoV-2 variants that were more common during the study period. The sample size of the panel was judged suitable in order to establish a sensitivity and specificity of >90% [9]. The multi-variant panel consisted of the following variants: Alpha (B.1.1.7) (*n* = 10), Beta (B.1.351) (*n* = 1), Gamma (P.1) (*n* = 12), Delta (B.1.617.2) (*n* = 32), Iota (B.1.526) (*n* = 5), Kappa (B.1.617.1) (*n* = 1) and wild-type strains (*n* = 11). The SARS-CoV-2 wild-type group was composed of strains belonging to B.1.177.15-Clade 20E (*n* = 1), B.1.258-Clade 20A (*n* = 2), B.1.160-Clade 20A (*n* = 1), B.1.177-Clade 20E (*n* = 6) and B.1.177.75-Clade 20E (*n* = 1).

### 2.1. Assays for SARS-CoV-2 Diagnosis

The Allplex SARS-CoV-2 Assay (Seegene Inc.; Seoul, South Korea) is a molecular test that was used for SARS-CoV-2 diagnosis of nasopharyngeal (NP) swabs, according to the manufacturer’s instructions. NP swabs were collected by using universal transport medium (Copan Diagnostics; Murrieta, CA, USA) or virus stabilization tube (Vacuette, Greiner Bio-One GmbH; Frickenhausen, Germany). The multiplex RT-PCR assay targets E, RdRp/S and N genes and was performed using the CFX96 instrument (Bio-Rad Laboratories, Hercules, CA, USA) following an extraction-free internal protocol [10]. Specimens with a cycle threshold (Ct) value <40 for at least two gene targets were defined positive.

### 2.2. Research Use Only (RUO) Assays for Screening of SARS-CoV-2 Variants

Five RUO multiplex real-time qualitative RT-PCR assays targeting some of the most widespread mutations in the spike protein (i.e., L452R, W152C, K417T, K417N, E484Q, E484K, N501Y) [11] were used on positive specimens for a rapid and presumptive detection of the SARS-CoV-2 variants (Figure 1 and Table 1). These molecular tests were performed on the automatically extracted RNA by means of the QIAamp Viral RNA Kits on QIAcube platform (QIAGEN Gmbh; Hilden, Germany) and stored at −20 °C. Specific RT-PCR procedures were conducted for each assay, according to the manufacturer’s instructions.

#### 2.2.1. SARS-CoV-2 Variants II Assay—Allplex (Seegene Inc.; Seoul, South Korea)

The SARS-CoV-2 Variants II Assay is a multiplex RT-PCR assay for the detection of spike protein mutations L452R, W152C, K417T and K417N, and it is validated for lower and upper respiratory tract specimens. Following the addition of the reaction components and extracted RNA in a single tube, the thermal profile was evaluated using the CFX96 instrument. Results were analyzed by means of proper viewer software, reporting data for each specific mutation probe as a Ct value.

#### 2.2.2. UltraGene Assay SARS-CoV-2 452R & 484K & 484Q Mutations V1.x (Advanced Biological Laboratories, ABL; Luxembourg, Luxembourg)

The UltraGene Assay SARS-CoV-2 452R & 484K & 484Q Mutations V1 is an RT-PCR test intended for use in the screening of SARS-CoV-2 mutations L452R, E484K and E484Q on the spike region from NP swab samples. Other specimens, such as oropharyngeal (throat) swabs, anterior nasal swabs, mid-turbinate nasal swabs, nasal aspirates, nasal washes and bronchoalveolar lavage fluid, are also reported by the manufacturer as acceptable specimen types for use. As recommended, samples suitable for the analysis must have a high viral load, determined by a Ct value ≤ 28. The thermal cycling program was operated using the Qiagen Rotor-Gene Q (QIAGEN Gmbh; Hilden, Germany). Detection of the amplicons for each specific SARS-CoV-2 target (L452R or E484) was performed using a single fluorescence channel, usually FAM. Analysis of the melting peak results for each mutation was conducted and is elaborated and reported in Appendix A.

#### 2.2.3. COVID-19 Ultra Variant Catcher (Clonit S.r.l.; Milan, Italy)

An alternative RUO assay for the identification of the SARS-CoV-2 variants is the COVID-19 Ultra Variant Catcher kit. L452R, E484K, E484Q and N501Y mutations are identified through this RT-PCR, usable for any type of positive SARS-CoV-2 biological samples. Reverse transcription and amplification were performed using the CFX96 thermal cycler to read FAM (wild-type sequences), HEX (E484K and E484Q mutations), Texas Red (N501Y mutation) and Cy5 (L452R mutation) fluorescence signals. The presence of the variants must be analyzed by means of Ct values for each specific probe, which detects the amino acid change and increases the amplification signal.

#### 2.2.4. SARS-CoV-2 Extended ELITe MGB (ELITechGroup; Puteaux, France)

This molecular kit is a multi-target designed assay intended for use as a reflex test for the detection of the mutations L452R, E484K, E484Q and N501Y of the spike gene on NP, oropharyngeal and nasal swabs. According to the manufacturer’s guidelines, following an RT-PCR, performed using the ELITe InGenius instrument (ELITechGroup; Puteaux, France), the identification of genic variations requires analysis of the melting curve (Appendix A).

#### 2.2.5. Simplexa SARS-CoV-2 Variants Direct (Diasorin Molecular; Vicenza, Italy)

Simplexa SARS-CoV-2 Variants Direct is an RT-PCR assay intended for detection of the N501Y, E484K, E484Q and L452 mutations in the spike genomic region of SARS-CoV-2 from NP or nasal swab samples from patients who are currently infected. This test requires the LIAISON MDX instrument, Direct Amplification Discs Kit and appropriate software for the analysis and interpretation of the results (LIAISON^®^ MDX Studio Software version 1.1 or higher). Following the reverse transcription, forward and reverse primers are used to amplify the SARS-CoV-2 S gene region containing the sites of mutations. Presence of mutations or wild-type sequences through the detection of amplicons with specific fluorescent probes (channels 520, 560, 610) is evaluated by melting curve analysis (Appendix A).

### 2.3. Interpretation of SARS-CoV-2 Variant Assays

Each assay can differentiate among some of these viral variants by identifying typical mutations: Alpha (B.1.1.7) by means of N501Y probe and, in some sequences, E484K probe; Beta (B.1.351) with N501Y, E484K and K417N probes; Gamma (P.1) with N501Y, E484K and K417T probes; Delta (B.1.617.2) through L452R probe and, in some sequences but not all, together with K417N probe; Iota (B.1.526) with E484K probe and, in some sequences, L452R probe; and Kappa (B.1.617.1) through L452R and E484Q probes. The absence of a fluorescence signal for these mutation probes with the detection of the peak only for endogenous gene, that is, ribonuclease P (RNase P) or generic RNA, as internal control or wild-type probes (i.e., HV69-70, E484 and N501 without mutations) can be defined as the identification of the viral strain without noted mutations, likely wild type.

Furthermore, these assays can be used to assume the presumptive presence of Eta (B.1.525) through E484K probe; Mu (B.1.6121) through N501Y and, occasionally, E484K; Zeta (P.1) through E484K; Theta (P.3) through E484K-N501Y; Epsilon (B.1.427/B.1.429) through L452 mutations; Lambda (C.37) through other L452 mutations; Omicron (B.1.1.529) through N501Y, K417N and E484, as well as others, that is, E484A; and B.1.640.2, which is the last discovered variant, often cited as “Cameroonian” (no WHO label has been attributed), through E484K and N501Y.

### 2.4. Genome Sequencing of SARS-CoV-2

Whole-genome sequencing or partial S gene sequencing of SARS-CoV-2 is the gold standard for identification of current and new variants. The library preparation for NGS was accomplished by using the multiplex or one-step RT-PCR methods through the CleanPlex SARS-CoV-2 FLEX Research and Surveillance Panel (Paragon Genomics, Inc.; Hayward, CA, USA) and AD4SEQ SARS-CoV-2 S gene (Arrow Diagnostics S.r.l.; Genoa, Italy) according to the manufacturer’s instructions. These amplicon-based SARS-CoV-2 NGS panels produce, respectively, whole-genome sequencing or only S gene sequencing of the virus. These panels were designed for research and surveillance purposes and require Illumina sequencing platforms.

Samples may undergo sequencing if their molecular assay readout for SARS-CoV-2 diagnosis shows Ct values ≤ 30; these results are used to determine the RNA input for the library.

Whole-genome sequencing was started by performing the synthesis of cDNA from the extracted RNA with the QIAamp Viral RNA Kit. Then, the multiplex PCR reaction with target-specific pools of primers to amplify targets of interest was completed. PCR products were then enzymatically digested to remove non-specific products, and a reaction using CleanPlex Indexed PCR primers was performed to amplify and add sample-level indexes to the generated libraries. All reactions, reverse-transcription, multiplex amplification, digestion and index PCR were performed following a purification phase by a magnetic bead-based clean-up step.

The final step of library preparation is quality control, performed by gel electrophoresis: a band at about 275 bp is associated with optimal amplification of the targeted regions. Single samples were also quantified by the Qubit 4 Fluorometer using the Qubit dsDNA HS Assay kit (Invitrogen, Thermo Fisher Scientific; Waltham, MA, USA).

The libraries were normalized using amplicon width (bp) and concentrations in order to make an equimolar pooling. Final pool was, again, quantified through the Qubit 4 Fluorometer, and 20 µL of the library (60 pM) was loaded into the cartridge introduced on the Illumina MiSeq platform (Illumina Inc.; Hayward, CA, USA). Cartridges were different, and their use depended on the available Illumina instrument and number of samples loaded for the sequencing.

The targeted S gene sequencing was characterized by a different workflow since the synthetized cDNA was amplified with specific target primers by a one-step PCR system. Following the bead-based clean-up and quality check, amplicons were used for the preparation of libraries through Illumina DNA Prep kit (Illumina Inc.; Hayward, CA, USA). The first step was tagmentation of the amplicons through the bead-linked transposomes to tagment DNA, which is a process that fragments and tags cDNA with adapter sequences. Then, the tagmented amplicons were indexed by PCR with IDT for Illumina DNA/RNA UD Indexes (Illumina Inc.; Hayward, CA, USA), following the bead-based clean-up. Prior to the final process steps, which are similar to the protocol steps reported above for the whole genome, the generated library was purified with a double bead-based clean-up in order to achieve a range of 400 bp.

Consensus sequences were elaborated using the SOPHia DDMTM platform (SOPHiA GENETICSTM Inc.; Boston, MA, USA) to align to the Wuhan-Hu-1 sequence and to detect all diverging gene mutations. Lineage and clade data, quality controls, mutations, missing nucleotides and gaps were analyzed and confirmed with the support of Pangolin (https://pangolin.cog-uk.io/ accessed on 15 November 2021) and Nextclade (https://clades.nextstrain.org/ accessed on 15 November 2021).

### 2.5. Data Collection and Statistical Analysis

Socio-demographic (age, sex) and clinical (comorbidities, COVID-19-related symptoms, duration of viral shedding, origin of case, first infection or reinfection, vaccination status, whether the case was imported and survival) data were collected. Disease severity was defined as mild (paucisymptomatic disease with fever lower than 37.5 °C for some days, mild cough and fatigue), moderate (common symptoms: fever higher than 37.8 °C, troublesome dry cough, fatigue; uncommon symptoms: diarrhea, headache, muscle pain, nausea, vomiting, chills, dizziness, loss of sense of smell and taste) or severe (breathless on light exertion, chest pain, loss of speech or movement).

Data are expressed as means with standard deviations (SDs) or medians with interquartile ranges (IQRs) for continuous variables and as proportions for categorical variables. Diagnostic performance characteristics of the assays used were assessed by calculating accuracy, specificity, sensitivity, Cohen’s κ, expected negative predictive values (NPVs) and positive predictive values (PPVs). The unpaired t test was used to compare continuous variables.

GraphPad Prism 8.0 version Software (GraphPad Software; San Diego, CA, USA) and Open Source Epidemiologic Statistics for Public Health (OpenEpi, https://www.openepi.com/ accessed on 15 November 2021) were used for the analysis. All results with *p*-values < 0.05 were considered statistically significant.

## 3. Results

### 3.1. Demographic and Clinical Characteristics of the Patients

From January to August 2021, a total of 302 SARS-CoV-2-positive samples were collected and tested by NGS during the monthly national “Flash Survey Variants”. Samples from patients with a history of vaccination or previous infection were also included in the sequencing analysis. Among them, the following VOCs were identified: 88 Alpha (B.1.1.7), 1 Beta (B.1.351), 17 Gamma (P.1 or sub-lineages P.1.1 and P.1.12) and 99 Delta (B.1.617.2 or sub-lineages AY.4/AY.9/AY.20/AY.25/AY.33/AY.34). VUMs accounted for only 5 Iota (B.1.526) and 1 Kappa (B.1.617.1) variants. Other or non-variants, likely wild-type strains, were detected for the remaining specimens sequenced. From this collection, we selected and analyzed a total of 72 upper respiratory tract samples for the multi-variant panel to be tested. Demographic and clinical features of the study participants are reported in Table 2.

Overall, the median age of patients was 36 (IQR: 23–55.5) years, and males and females were equally distributed (36 of 72, 50%). Comorbidities were observed in 48 (66.6%) patients; most of these were affected by diabetes (12 of 48, 25%) and cardiovascular disease (9 of 48, 19%). Other comorbidities included human immunodeficiency virus infection (1 of 48, 2%), autoimmune disease (1 of 48, 2%), hypertension (1 of 48, 2%) and chronic respiratory disease (1 of 48, 2%). Clinical data were not reported for the remaining patients with comorbidities (22 of 48, 48%). No reinfection or imported cases were registered. The origin of cases was categorized as screening, contact tracing and symptomatic patients. Out of 72 cases, 8 (11%) were detected by screening and 10 (14%) by contact-tracing activities, and the remaining 54 (75%) were patients with symptoms. The overall survival was 98.6%, with a total follow-up period of 112 ± 70 days. The only fatal event concerned an 87-year-old woman with a history of cancer; she was infected with the Alpha VOC and was not vaccinated. Mild (62.5%), moderate (11%) and severe (5.5%) symptoms were observed in 57 (77.7%) patients. Among asymptomatic patients, similar proportions of Alpha (B.1.1.7) (5 of 16, 26.6%), Delta (B.1.617.2) (4 of 16, 26.6%) and wild-type strains (4 of 16, 26.6%) were observed. Analogously, for this patient category, Gamma (P.1), Iota (B.1.526) and Kappa (B.1.617.1) represent the minority.

The mean duration of SARS-CoV-2 shedding was 17 ± 8 days (maximum 59 days). As expected, an increased duration of viral shedding was observed in patients with severe symptoms (34 ± 17 days).

There were 9 (12.5%) fully vaccinated patients, of which 67% (6 of 9) were female, and their median age was 41 (IQR: 32–54) years. Most of these patients received a double dose of the BNT162b2 mRNA vaccine, and only one patient was immunized with the single-dose Ad26.COV2-S vaccine. Overall, the mean time from vaccine efficacy, which is reached one week after the second dose, to infection was 115 ± 72 days (maximum of 200 days). The mean duration of SARS-CoV-2 RNA shedding for these cases (18 ± 7 days) did not reduce, despite the immunological protection. However, by excluding high-risk patients aged >50 years [12,13], the mean duration of SARS-CoV-2 RNA shedding decreased remarkably (13 ± 1 days). Similar results were obtained for the total study population: patients aged >50 years and those aged <50 years had the mean duration of SARS-CoV-2 RNA shedding of 20 ± 11 days and 15 ± 5 days, respectively. The SARS-CoV-2 strain of vaccinated patients was Delta (B.1.617.2) in eight (89%) cases, while the wild-type (B.1.177) strain was detected in only one (11%) patient. This finding was expected since the Delta SARS-CoV-2 VOC was the most diffused globally during and after the vaccination campaign.

### 3.2. Accuracy, Sensitivity and Specificity of Real-Time RT-PCR SARS-CoV-2 Variant Assays

The main diagnostic performance characteristics of the assays analyzed are reported in Table 3. The accuracy among the different RUO assays ranged from 96.9% (95% CI: 89.6–99.1%) to 100% (95% CI: 95–100%). The overall specificity was 100% (95% CI: 74.1–100%) and the sensitivity was 100% (95% CI: 94.1–100%) for the SARS-CoV-2 Variants II Assay, COVID-19 Ultra Variant Catcher and SARS-CoV-2 Extended ELITe MGB kits. The UltraGene Assay SARS-CoV-2 452R & 484K & 484Q Mutations V1 and Simplexa SARS-CoV-2 Variants Direct kits showed a sensitivity of about 96% (both detected two mutations not confirmed by the NGS). In particular, in the former assay, L452R mutations were observed for two samples positive for E484K. In the latter assay, one sample was identified as wild type instead of Delta (B.1.617.2) for the L452R mutation, and in another sample, Gamma (P.1) was only positive for N501Y with a drop-out for the E484K mutation. All tests showed an NPV between 84.6% and 100% and a PPV of 100%.

The overall mean Ct value was 23.6 ± 3.8 and without any statistically significant difference (*p* = 0.641) between samples detected correctly (23.6 ± 3.8) and those identified incorrectly (22.7 ± 5.7).

All sequences under analyses showed approximately 90% of the genome or targeted S gene coverage.

## 4. Discussion

Sequencing is the reference method used to monitor the diffusion of emerging SARS-CoV-2 variants via the Sanger or NGS method. However, these techniques are characterized by high costs and a relatively long turnaround time (TAT) of about 3–4 days, affected by working sessions with low a volume of samples, require specific laboratory equipment and instruments and, therefore, may only be performed by few specialized diagnostic centers. A prompt and widespread epidemiological surveillance of COVID-19 is not feasible using only these methods. To optimize control of SARS-CoV-2 variants, the molecular detection of the most widespread spike protein mutations could be used as a test of the second level in the diagnostic workflow of COVID-19 (Figure 2).

In this study, we tested five RUO real-time RT-PCR assays for the presumptive detection of SARS-CoV-2 VOCs and VUMs, and all kits tested were observed to be rapid, cheap and useful techniques in the routine of a COVID-19 laboratory. The SARS-CoV-2 Variants II Assay, COVID-19 Ultra Variant Catcher kit and SARS-CoV-2 Extended ELITe MGB Kit reached a maximum level of accuracy and confirmed all the mutations detected by sequencing. However, the UltraGene Assay SARS-CoV-2 452R & 484K & 484Q Mutations V1 and Simplexa SARS-CoV-2 Variants Direct also displayed good performance, with an accuracy of 97.2% and 96.9%, respectively. An additional diagnostic feature to evaluate these tests is the different number of detected mutations. All assays detect four different mutations, except for the Ultra Variant Catcher kit, which targets the E484 position without distinguishing between E484K and E484Q, and the UltraGene Assay SARS-CoV-2 452R & 484K & 484Q Mutations V1, which is designed for three mutation targets. SARS-CoV-2 Variants I and II assays have already been analyzed by some studies conducted between November 2020 and June 2021. These studies defined these assays as feasible and user friendly and encouraged their clinical application as second-line mutation screening assays to monitor local epidemiology [14,15,16]. Moreover, several studies reported similar satisfying results by using in-house molecular tests for variant screening of both clinical and environmental specimens. By allowing a rapid and accurate detection of mutations, these studies have highlighted the importance of focusing resources of genome sequencing on the research of newly emerging variants [17,18,19,20]. To date, no data on the diagnostic performance of the other assays analyzed in the present study are available.

It is known that a low viral load is the main factor of discrepant results between sequencing and screening tests for SARS-CoV-2 variants [21]. Indeed, in this study, all samples tested had a high viral load (mean Ct value of 23.6 ± 3.8) in order to guarantee that the diagnostic evaluation of the assays was performed on good-quality samples. Furthermore, our library preparation procedures required that samples have Ct ≤ 30. Due to these reasons, we were not able to evaluate the diagnostic performance of the assays tested in a wider range of viral loads.

In this study, we also analyzed the main demographic and clinical data of the patients included in the selection of the multi-variant panel. Comorbidities, such as diabetes (25%) and cardiovascular disease (19%); mild symptoms, such as fever, cough and fatigue (62.5%); and mortality rate (1.4%) were in line with the data reported in the available clinical studies [22,23,24,25]. Despite the fact that the samples analyzed were collected during 2021 when the diffusion of VOCs was higher, we did not find evidence of an increase in symptoms, mortality or the duration of viral shedding [22,23,24,25,26,27].

The analyzed real-time RT-PCR assays target three or four of the following amino acid variations: L452R, W152C, K417T, K417N, E484Q, E484K and N501Y. These single mutations are shared by two or more variants, except for W152C, which is described only for Epsilon (B.1.427 and B.1.429 lineages). The presence of one or two of these mutations can often be presumptive of one or more variants. Therefore, to increase the benefit of these assays in epidemiological surveillance, each single mutation, or its combination with other mutations, should be interpreted with caution and with consideration of the circulating variants.

In addition, it could be advantageous to design a reflex molecular kit to screen variants with different multiplex RT-PCR assays. The first assay should detect some common mutations of the circulating variants. The obtained results should drive the selection of the second RT-PCR assay targeting other mutations for the final interpretation of the presumptive variant. Despite the prolonged TAT, this strategy may be able to discriminate a single variant from others during the periods characterized by the circulation of more variants sharing the same mutations.

Furthermore, these assays must be under continuous development due to the fact that SARS-CoV-2 continuously acquires novel mutations. For example, nowadays, the diffusion of the new Omicron variant can be quickly monitored by the means of a test identifying one or more of its new and unshared spike mutations (i.e., ins214EPE, S371L and S373P). However, the use of the SARS-CoV-2 Variants II Assay—Allplex for K417N and the COVID-19 Ultra Variant Catcher, SARS-CoV-2 Extended ELITe MGB or Simplexa SARS-CoV-2 Variants Direct for N501Y and E484 mutations can discriminate the Omicron variant from the Delta variant. At the same time, they could be performed for the presumptive detection of the recent “Cameroonian” variant, which is B.1.640.2 (to date, no WHO label has been attributed). Therefore, despite the fact that these different screening assays for SARS-CoV-2 variants were produced in 2020, they continue to be useful in the new pandemic scenario characterized by the increasing circulation of the Omicron variant.

The clinical use of these screening tests, especially when viral transmission increases, may accelerate surveillance activities. Sequencing could be performed on wild-type SARS-CoV-2 samples for molecular variant assays or with discordant mutations and mostly for the research of newly emerging variants.

This study is characterized by some limitations. First, the sample size is limited by the number of available tests. Moreover, reproducibility and cross-reactivity were not tested for the same reason. Second, the analyzed multi-variant panel is skewed toward Delta and Alpha variants because, during the study period, the incidence of other variants and wild-type strains was low. However, we believe that these limitations do not affect the study results.

## 5. Conclusions

The molecular SARS-CoV-2 variant assays analyzed in this study are simple, fast and low cost. In view of these advantages and the high level of accuracy demonstrated in the analysis, we highly recommend their use in the workflow of COVID-19 diagnostic laboratories as second-level tests before the implementation of sequencing. These tests identify specific mutations in about 2 h–3 h, accelerating the surveillance activity of VUMs and VOCs.

## Figures and Tables

**Figure 1 microorganisms-10-00306-f001:**
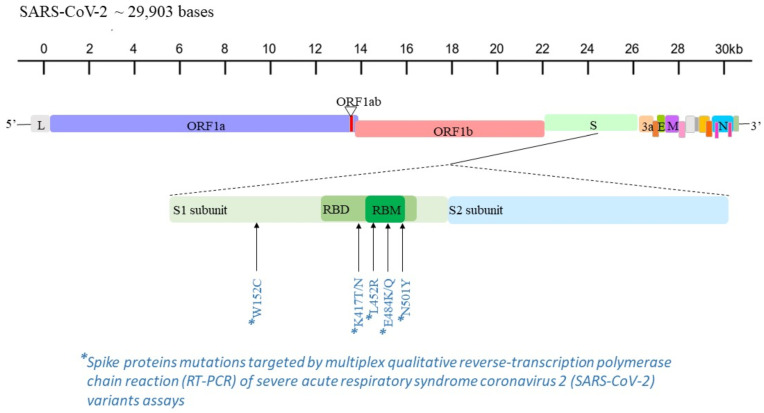
Mapping of L452R, W152C, K417T, K417N, E484Q, E484K and N501Y mutations in the genome sequence of SARS-CoV-2 [11].

**Figure 2 microorganisms-10-00306-f002:**
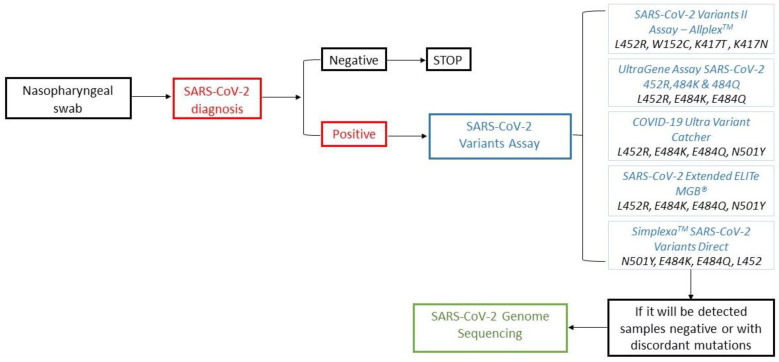
Workflow of SARS-CoV-2 diagnosis in the era of variants of concern (VOCs) and variants under monitoring (VUMs).

**Table 1 microorganisms-10-00306-t001:** Spike protein mutations identifiable by the SARS-CoV-2 variant assays used.

SARS-CoV-2 Variants Assays	Mutation
SARS-CoV-2 Variants II Assay	L452R, W152C, K417T, K417N
UltraGene Assay SARS-CoV-2 452R & 484K & 484Q Mutations V1	L452R, E484K, E484Q
COVID-19 Ultra Variant Catcher kit	L452R, E484K, E484Q, N501Y
SARS-CoV-2 Extended ELITe MGB Kit	L452R, E484K, E484Q, N501Y
Simplexa SARS-CoV-2 Variants Direct	N501Y, E484K, E484Q, L452

**Table 2 microorganisms-10-00306-t002:** Demographic and clinical characteristics of the study patients (*N* = 72).

Male, N (%)	36 (50%)
Age (yr), median (IQR)	36 (23–55)
Comorbidities, N (%)	48 (66.6%)
Diabetes	12 (25%)
Cardiovascular disease	9 (19%)
Origin of case, N (%)	
Screening	8 (11%)
Contact tracing	10 (14%)
Symptomatic patients	54 (75%)
COVID-19-related symptoms, N (%)	
None	15 (21%)
Mild	45 (62.5%)
Moderate	8 (11%)
Severe	4 (5.5%)
SARS-CoV-2 shedding duration (days), mean (SD)	17 ± 8 days
Vaccinated patients, N (%)	9 (12.5%)

**Table 3 microorganisms-10-00306-t003:** Retrospective comparison of real-time RT-PCR SARS-CoV-2 variant assays and genome sequencing.

SARS-CoV-2 Variant Assays	Sensitivity	Specificity	Positive Predictive Values (PPVs)	Negative Predictive Values (NPVs)	Accuracy	Cohen’s k
SARS-CoV-2 Variants II Assay	100% (94.1–100%)	100% (74.1–100%)	100% (94.1–100%)	100% (74.1–100%)	100% (95–100%)	1 (0.77–1)
UltraGene Assay SARS-CoV-2 452R & 484K & 484Q Mutations V1	96.7% (88.9–99.1%)	100% (74.1–100%)	100% (93.9–100%)	84.6% (57.7–95.6%)	97.2% (90.5–99.2)	0.90 (0.67–1)
COVID-19 Ultra Variant Catcher kit	100% (94.1–100%)	100% (74.1–100%)	100% (94.1–100%)	100% (74.1–100%)	100% (95–100%)	1 (0.77–1)
SARS-CoV-2 Extended ELITe MGB Kit	100% (94–100%)	100% (74.1–100%)	100% (94–100%)	100% (74.1–100%)	100% (94.9–100%)	1 (0.76–1)
Simplexa SARS-CoV-2 Variants Direct	96.3% (87.6–99%)	100% (74.1–100%)	100% (93.2–100%)	84.6% (57.7–95.6%)	96.9% (89.6–99.1%)	0.89 (0.65–1)

Data were analyzed using Open Source Epidemiologic Statistics for Public Health (OpenEpi, https://www.openepi.com/ accessed on 15 November 2021) and reported as estimate (95% CI).

## Data Availability

The datasets used and/or analyzed during the current study are available from the corresponding author on reasonable request.

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
