# Peer review of "Comparative Analysis of Five Multiplex RT-PCR Assays in the Screening of SARS-CoV-2 Variants"

_microorganisms, 2022, doi:10.3390/microorganisms10020306_

Round 1
Reviewer 1 Report
In this manuscript Pace et al., evaluated and compared five research use only (RUO) real time RT-qPCR assays for presumptive detection of SARS-CoV-2 variants. A total of 72 COVID-19 positive samples were sequenced using illumina platform for identification of variants, and different variants were subsequently screened by RT-qPCR. These variants include SARS-CoV-2 variants of concern, variants of interest, and variants under monitored - categorized based on disease severity, viral phenotypic and epidemiological characteristics, etc. (published reports). However, there were only one sample evaluated in the screening assays for the detection of Beta (B.1.351) and Kappa (B.1.617.1) variants. This study addresses the diagnostic performances of assays for the current pandemic, and should be of interest to clinicians, epidemiologists, and other scientists. I have few concerns that needs to be discussed to the study findings.
Major comments:
All samples evaluated in this study contains high viral load, the diagnostic assay (evaluated among 5 RUO) that can be realistically used for identification of COVID-19 variants with low viral load remains unknown. This limitation in evaluating the analytical sensitivity should be appropriately discussed. Though the library preparation and NGS criteria required samples of Ct ≤30, the authors could have attempted to perform dilutions and tested low ct values to determine the assay performance characteristics.
Minor concerns:
- line 147 Should indicate the internal control target here or in methods.
- Figs 1 and 2. Font size is small, and not clear. This needs to be revised.
- lines 380-384 not clear.
- 72 positive samples were analyzed. It is not clear how specificity is evaluated?
- The five RUO assays also targets detection of different mutations, for eg. UltraGene Assay SARS-CoV-2 detects 452R & 484K & 484Q mutations, vs. COVID-19 Ultra Variant Catcher detects an additional N501Y mutation. The significance of assays in detecting additional mutations over the other assay could be included in discussion lines 344-355 - where the authors illustrate the assay performance characteristics.
Author Response
Author reply Manuscript ID: microorganisms-1551040
Comparative analysis of five multiplex RT-PCR assays
in the screening of SARS-CoV-2 Variants
Minor Revisions
Reviewer 1
Comments and Suggestions for Authors
In this manuscript Pace et al., evaluated and compared five research use only (RUO) real time RT-qPCR assays for presumptive detection of SARS-CoV-2 variants. A total of 72 COVID-19 positive samples were sequenced using illumina platform for identification of variants, and different variants were subsequently screened by RT-qPCR. These variants include SARS-CoV-2 variants of concern, variants of interest, and variants under monitored - categorized based on disease severity, viral phenotypic and epidemiological characteristics, etc. (published reports). However, there were only one sample evaluated in the screening assays for the detection of Beta (B.1.351) and Kappa (B.1.617.1) variants. This study addresses the diagnostic performances of assays for the current pandemic, and should be of interest to clinicians, epidemiologists, and other scientists. I have few concerns that needs to be discussed to the study findings.
Author reply (AR)
We are very grateful with the Reviewer to highlight the interest of this study for clinicians and epidemiologists and other physicians involved in the management of this current pandemic.
As reported in the Results lines 238-241 during monthly national Flash Survey Variants, Beta (B.1.351) and Kappa (B.1.617.1) variants as well as Iota (B.1.526) are low representative, as well as in the multi-variants panel under analysis, because in Liguria region their circulation in the study period were very low. The number of the cases included reflects those sequenced from January to August 2021.
Major comments:
All samples evaluated in this study contains high viral load, the diagnostic assay (evaluated among 5 RUO) that can be realistically used for identification of COVID-19 variants with low viral load remains unknown. This limitation in evaluating the analytical sensitivity should be appropriately discussed. Though the library preparation and NGS criteria required samples of Ct ≤30, the authors could have attempted to perform dilutions and tested low ct values to determine the assay performance characteristics.
AR
We appreciate the Reviewer for this comment. In lines 352-358, we report the response about this question. Migueres M. and collegues documented that the most of the discrepant results between sequencing and multiple RT-PCR assays targeting single nucleotide polymorphism concerned samples with low viral loads. In addition, in the Instructions for use of some tests (i.e. UltraGene Assay SARS-CoV-2 452R & 484K & 484Q Mutations V1.x), it is recommended the use of samples with high viral load, i.e. Ct value ≤ 28 as reported in the text (lines 116-117). Despite these observations, it will be interesting to evaluate the analytical sensitivity of our RUO assays. However, as reported in the discussion lines 397-399, the number's available tests of each molecular procedure here examined, has limited the analysis of this diagnostic characteristic.
Minor concerns:
line 147 Should indicate the internal control target here or in methods.
Figs 1 and 2. Font size is small, and not clear. This needs to be revised.
lines 380-384 not clear.
72 positive samples were analyzed. It is not clear how specificity is evaluated?
The five RUO assays also targets detection of different mutations, for eg. UltraGene Assay SARS-CoV-2 detects 452R & 484K & 484Q mutations, vs. COVID-19 Ultra Variant Catcher detects an additional N501Y mutation. The significance of assays in detecting additional mutations over the other assay could be included in discussion lines 344-355 - where the authors illustrate the assay performance characteristics.
AR
As requested by the Reviewer, we revised the following points:
- In the lines 156-160 it was reported the internal control or wild type probes of the reference methods.
- Font size of the Figure 1 and 2 was modified, as required.
- We reformulate the concept expressed in these lines, as suggested by the Reviewer (lines 374-380).
- The study was focused on SARS-CoV-2 Variants and multiplex RT-PCR assays intended to use for qualitative detection of specified Spike protein mutations. Therefore, a negative result for these assays is characterized by none signals of amplification for the mutation targets. Specificity was calculated elaborating data of the wild type group.
- An additional comment about the different number of mutations targeting by all assays, it was included in this paragraph.
The lines cited in these responses are referred to the manuscript version without track changes.

Reviewer 2 Report
MICROORGANISMS - MDPI
Manuscript ID: 1551040
Thank you very much for the opportunity given to me to review the manuscript entitled “Comparative analysis of five multiplex real-time RT-qPCR assays in the screening of SARS-CoV-2 Variants” by De Pace et al. that I found interesting to screen different, but not all, genotypes of circulating SARS-CoV-2.
The research question is clearly stated and the mythology is well stated, however it is open to criticism that this manuscript described old genotypes of SARS-CoV-2, which are not circulating nowadays. Moreover, there are some points that need revision:
TITLE
- As only some variants of this virus are susceptible to be screened with one or all these five assays, the use of the words “screening of SARS-CoV-2” is confusing. The title should mentioned which variants are screened.
ABSTRACT
- Major critic = line 16: why only 72 samples were tested for a variety of variants? How many was their proportion regarding to the total positives in your country? Is this number is representative of the pandemic and the number of the positive cases between January and August 2021?
- Lines 16 to 18: The date of categorization of the variants as VUM and/or VOC must be mentioned. For example, Alpha variant was considered, as you know and well described in the Introduction section, as VOC at the beginning of the year 2021, but the authors mention it as non-VOC in this article.
- Major critic = All these variants (except Delta) are not circulating nowadays in Europe. What about the Omicron variant?
INTRODUCTION
- Lines 63: Discrepancy in the months of the survey. While it was cited “between February and August” herein, it was mentioned between January and August in the Abstract section, why?
- Line 68: RNA was extracted, not samples were extracted. Please, correct.
MATERIALS AND METHODS
- Major critic = lines 72 to 76: the explanation given to choose only 72 samples to be tested seems to be wrong because the specificity and the sensitivity are completely independent of the number of tests. Moreover, how many was the proportion of the 72 samples regarding to the total positives in your country? This number is not representative of the pandemic and the huge number of the positive cases between January/February and August 2021.
- Major critic = line 85: Could the authors explain how can a Ct value between 36 and 39.99 could be considered as positive, please justify and discuss OR exclude them from this study?
- Figure 1: the abbreviation need to be detailed, please.
- Lines starting from 186: how many of the 72 were partially sequenced for the S-gene only?
RESULTS
- Table 1 – Text: Discrepancy between the reported number of male between the table (36) versus the text (31)? Which one is correct, please?
- Lines 236-237: please give the exact number of each, because “were rarely observed” is not clear.
- Lines 244-245: Another discrepancy between the data in the text and in the table, Low+mild+severe = 57 not 56 (in the text), please verify which is correct?
- Lines 252-265: Could the authors give the age and sexe of the vaccinated persons that were infected after their vaccination?
- Line 263 and the whole text: what is the wild type of SARS-CoV-2?
DISCUSSION
- Line 376: please separate the two names of Epsilon variant.
SUPPLEMENTARY TABLES
- The informations reported by these Tables are very interesting, which could suggest putting them in the main text, if possible.
Author Response
Author reply Manuscript ID: microorganisms-1551040
Comparative analysis of five multiplex RT-PCR assays
in the screening of SARS-CoV-2 Variants
Minor Revisions
Reviewer 2
Thank you very much for the opportunity given to me to review the manuscript entitled “Comparative analysis of five multiplex real-time RT-qPCR assays in the screening of SARS-CoV-2 Variants” by De Pace et al. that I found interesting to screen different, but not all, genotypes of circulating SARS-CoV-2.
The research question is clearly stated and the mythology is well stated, however it is open to criticism that this manuscript described old genotypes of SARS-CoV-2, which are not circulating nowadays. Moreover, there are some points that need revision:
- Title: As only some variants of this virus are susceptible to be screened with one or all these five assays, the use of the words “screening of SARS-CoV-2” is confusing. The title should mentioned which variants are screened.
AR
We appreciate the Reviewer’s opinion and we are grateful for his interest about this research context. Last SARS-CoV-2 variants, that is Omicron, could be similarly identified by means of these tests because it is characterized in the Spike region by K417N, E484- and N501Y. In addition, some of these tests can discriminate in the ongoing pandemic wave Delta with L452R mutation from Omicron. This issue was discussed in the first submitted manuscript, now we have added some comment.
The presumptive variants of SARS-CoV-2 that they can be investigate by means of these tests, as reported in the “Interpretation of SARS-CoV-2 variants assays” section, are several for the inclusion in the title.
ABSTRACT
Major critic = line 16: why only 72 samples were tested for a variety of variants? How many was their proportion regarding to the total positives in your country? Is this number is representative of the pandemic and the number of the positive cases between January and August 2021?
AR
As reported in the Discussion lines 397-398 about the study limitations, it was cited the low sample size because there were a limited available number of tests. However, the number of the cases included reflects those identified and sequenced from January to August 2021. Indeed, some variants, i.e. Beta (B.1.351), Kappa (B.1.617.1) and Iota (B.1.526), are low representative in the multi-variants panel under analysis because in Liguria region their circulation in the study period were very low (as reported in the lines 235-242).
Lines 16 to 18: The date of categorization of the variants as VUM and/or VOC must be mentioned. For example, Alpha variant was considered, as you know and well described in the Introduction section, as VOC at the beginning of the year 2021, but the authors mention it as non-VOC in this article.
AR
In the introduction (lines 32-45) as well as in the results sections (lines 235-242), we report the current interpretation by WHO of VOC, VUM and VOI. Alpha variant is mentioned among VOCs.
The date of access online at World Health Organization - Tracking SARS-CoV-2 variants, https://www.who.int/en/activities/tracking-SARS-CoV-2-variants/, is 26 November 2021 and it was included as required.
Major critic = All these variants (except Delta) are not circulating nowadays in Europe. What about the Omicron variant?
AR
As reported above and in the lines 164-165, now high spread SARS-CoV-2 variant, that is Omicron, can be similarly identified by means of these tests because it is characterized in the Spike region by K417N, E484- and N501Y. At the same time, the additional detection of L452R with only one assay can be discriminates Delta from Omicron.
This study was write during the third pandemic wave when Delta was the only variant circulating. Therefore, now we added in the line 165 the potential detection of B.1.640.2 Cameroonian variant with the same assays and we discussed in the lines 381-392 about the new circulating variants that these assays can identify, that they are B.1.529 (Omicron) and B.1.640.2 (none label WHO).
INTRODUCTION
- Lines 63: Discrepancy in the months of the survey. While it was cited “between February and August” herein, it was mentioned between January and August in the Abstract section, why?
- Line 68: RNA was extracted, not samples were extracted. Please, correct.
AR
We thank the Reviewer for these important comments and we report that it was modified in line 63 the correct study period, that is from January to August 2021, and the “RNA extraction” sentence in the lines 67 and 94.
MATERIALS AND METHODS
Major critic = lines 72 to 76: the explanation given to choose only 72 samples to be tested seems to be wrong because the specificity and the sensitivity are completely independent of the number of tests. Moreover, how many was the proportion of the 72 samples regarding to the total positives in your country? This number is not representative of the pandemic and the huge number of the positive cases between January/February and August 2021.
AR
We thank the Reviewer for these important questions. In the lines 72-74, it was affirmed only that sample size used was suitable to analyze sensitivity and specificity of the selected assays. However, we modified this sentence to clarify it.
Instead, as reported in the Discussion (lines 397-398) about the study limitations, it was cited the low sample size because there were a limited available number of tests. However, the number of the cases included reflects those identified and sequenced from January to August 2021. Indeed, some variants, i.e. Beta (B.1.351), Kappa (B.1.617.1) and Iota (B.1.526), are low representative in the multi-variants panel under analysis because in Liguria region their circulation in the study period were very low (as reported in the lines 235-245).
In addition, we report that, during each Quick Survey SARS-CoV-2 Variants supported by the Italian National Institute of Health and Ministry of Health, it was calculated the number of sequences to elaborate by every participating lab. This sample volume was defined in order of the number of notified cases in the week before the survey, assuming we want to estimate a prevalence of 5% with 2% accuracy.
Major critic = line 85: Could the authors explain how can a Ct value between 36 and 39.99 could be considered as positive, please justify and discuss OR exclude them from this study?
AR
In the lines 84-85, we report that “Specimens were defined as positive with a cycle threshold (Ct) value <40 for at least two gene targets.” with the aim to clear for all experts of general laboratory medicine but not of SARS-CoV-2 diagnosis how to interpret a positive result. Ct values between 36 and 40 for 2 of 3 gene targets could be interpreted as weakly positive but this evaluation requests clinical and age data as well as a specified history of each study case. However, we consider that this study is not focused on the interpretation of the results of SARS-CoV-2 diagnosis. For this reason, it was not specified between positive or weakly positive in this sentence.
In adding, in the lines 176-177, we write this sentence: “Samples are useful to sequence if their molecular assay for SARS-CoV-2 diagnosis show Ct ≤30”. As well as, in the lines 117, “As recommended, samples suitable for the analysis must have high viral load, that is a Ct value ≤ 28.”.
Figure 1: the abbreviation need to be detailed, please.
AR
As required, we have detailed the abbreviations in the figure 1.
Lines starting from 186: how many of the 72 were partially sequenced for the S-gene only?
AR
As suggested, we have added in this line the number of full genome (N = 69) and S gene (N = 3) sequences.
RESULTS
- Table 1 – Text: Discrepancy between the reported number of male between the table (36) versus the text (31)? Which one is correct, please?
- Lines 236-237: please give the exact number of each, because “were rarely observed” is not clear.
- Lines 244-245: Another discrepancy between the data in the text and in the table, Low+mild+severe = 57 not 56 (in the text), please verify which is correct?
- Lines 252-265: Could the authors give the age and sexe of the vaccinated persons that were infected after their vaccination?
- Line 263 and the whole text: what is the wild type of SARS-CoV-2?
AR
- Because it was a writing mistake, we modified the number of male sex (50%) cases in our population (N=72), that is 36.
- As required, we report in the text the number of cases of the following comorbidities: human immunodeficiency virus infection (N = 1), autoimmune disease (N = 1), hypertension (N = 1), chronic respiratory disease (N = 1). In addition, we report that 48% of comorbidities were not specified in the available clinical data.
- As correctly noted, we modified this calculation mistake.
- As required by the Reviewer, we included in the lines 267-268 the sex and age data of the vaccinated study cases.
- SARS-CoV-2 wild type group is characterized by all viruses which they are not viral variants and they were circulating in the same period of viral variants under analyses. These strains are missing of the Spike mutations of interest. It is composed by: B.1.177.15 - Clade 20E (N = 1), B.1.258 - Clade 20A (N = 2), B.1.160 - Clade 20A (N = 1), B.1.177 - Clade 20E (N = 6), B.1.177.75 - Clade 20E (N = 1). We have added in the text (line 278) the lineage data of the study case that is wild-type and with vaccination.
DISCUSSION
- Line 376: please separate the two names of Epsilon variant.
AR
We thank the Reviewer for this correction; we modified as noted.
SUPPLEMENTARY TABLES
- The informations reported by these Tables are very interesting, which could suggest putting them in the main text, if possible.
AR
We appreciate this Reviewer’s suggestion. However, these tables were included in the Supplemental Material for space reasons. If possible, we will available to modify their placement.
The lines cited in these responses are referred to the manuscript version without track changes.

Reviewer 3 Report
De Pace et al present “Comparative analysis of five multiplex real-time RT-qPCR assays in the screening of SARS-CoV-2 Variants”, a manuscript aimed at evaluating commercial kits for the classification of SARS-CoV-2 variants detected from nasopharyngeal swabs. This is an interesting study, and ways to cut costs are desperately needed in the detection of SARS-CoV-2 variants, especially as we move towards a virus becoming seasonal.
General comments:
These RUO assays may not be useful to detect new variants, in comparison to NGS – this should be discussed
I would suggest to the authors to use “RT-qPCR” instead of using the words “real time” as these can sometime get mixed up with Reverse Transcription as the RT.
Line by line comments:
Line 17-18 “in our region” change to “in Liguria, Italy” for example, please include the information
Line 19 – these were mapped to Wuhan isolate?
Line 24, who is highly recommending these tests
Line 35 “variants under monitoring”
Line 54, syntax problem, suggesting SARS-CoV-2 has a diagnostic activity, this needs rewording
Line 56, COVID-19 should be first defined in line 31 where it is first used
Line 64 “SARS-CoV-2 positive samples”
Line 69-70, would this study have benefitted from blinding, as tests were carried out after NGS? Please could the authors discuss or reply to this
Line 76, the authors should define “wild type”, in this case being B lineage from Wuhan?
Line 77 and 86, a small table of the used tests would be helpful
Line 86 onwards, a table highlighting the coverage of each RUO would be useful (including which mutations are detected, etc), this could be combined with the above comment
Line 210, a citation for WHO guidelines would be useful
Line 234, 31/72 is supposed to be 43% ?
Line 249-251, clarity needed here, was a patient tested daily for 59 days? And other patients too, with the majority testing positive by PCR for 17 +/- 8 days?
Line 254 “achievement of vaccine efficacy” needs rewording, or clarification, do the authors mean the time from first (or full) vaccination to infection?
Line 258, why was the age of 50 used as a cut-off, are there references to cite for this decision?
Line 267, mention of table 2, should this be before?
Line 274, two wrong mutations when compared to NGS data?
Line 288, figure 1, please increase the text resolution of the mutations
Line 330, increase text resolution in fig 2 please
Line 330, it was previously reported that NGS occurred before RUO use, please clarify which way it is! The figure suggests RUO is carried out before NGS
Line 345, the authors have not stated the costs of these tests, aside from terming them “cheap”,
Author Response
Author reply Manuscript ID: microorganisms-1551040
Comparative analysis of five multiplex RT-PCR assays
in the screening of SARS-CoV-2 Variants
Minor Revisions
Reviewer 3
Comments and Suggestions for Authors
De Pace et al present “Comparative analysis of five multiplex real-time RT-qPCR assays in the screening of SARS-CoV-2 Variants”, a manuscript aimed at evaluating commercial kits for the classification of SARS-CoV-2 variants detected from nasopharyngeal swabs. This is an interesting study, and ways to cut costs are desperately needed in the detection of SARS-CoV-2 variants, especially as we move towards a virus becoming seasonal.
General comments:
These RUO assays may not be useful to detect new variants, in comparison to NGS – this should be discussed
AR
We are very grateful with the Reviewer for this opinion. Last SARS-CoV-2 variants, that is Omicron, could be similarly identified by means of these tests because it is characterized in the Spike region by K417N, E484- and N501Y. In addition, some of these tests can discriminate in the ongoing pandemic wave Delta with L452R mutation from Omicron. Indeed, as reported in the “Interpretation of SARS-CoV-2 variants assays” section, these assays can be used also to assume the presumptive presence of Omicron (B.1.1.529) for N501Y, K417N and E484 other that is E484A, and B.1.640.2 for E484K and N501Y, that is the last discover variant cited as Cameroonian (none WHO label). Furthermore, in the discussion we have added from lines 381-392 a brief discussion about these new additional variants which these assays can detect in the current pandemic wave.
I would suggest to the authors to use “RT-qPCR” instead of using the words “real time” as these can sometime get mixed up with Reverse Transcription as the RT.
AR
We modified all abbreviations RT-qPCR with RT-PCR because it is a real-time qualitative reverse transcription (RT) polymerase chain reaction (PCR), RT-PCR, and we extended at the first citation this abbreviation.
Line by line comments:
Line 17-18 “in our region” change to “in Liguria, Italy” for example, please include the information
AR
As required, we modified this expression.
Line 19 – these were mapped to Wuhan isolate?
AR
All sequences were aligned to the Wuhan-Hu-1 sequence using SOPHia DDMTM platform SOPHiA GENETICSTM Inc.; Boston, USA) and now, it was specified in the lines 209-211.
Line 24, who is highly recommending these tests
AR
We modified this sentence in the abstract and in the conclusion.
Line 35 “variants under monitoring”
AR
Variants Under Monitoring (VUM) is defined by World Health Organization as “SARS-CoV-2 variant with genetic changes that are suspected to affect virus characteristics with some indication that it may pose a future risk, but evidence of phenotypic or epidemiological impact is currently unclear, requiring enhanced monitoring and repeat assessment pending new evidence.” Link at https://www.who.int/en/activities/tracking-SARS-CoV-2-variants/.
Line 54, syntax problem, suggesting SARS-CoV-2 has a diagnostic activity, this needs rewording
AR
As suggested, we re-write this sentence as follows: “These variants have increased the diagnostic activity for SARS-CoV-2, which is committed to diagnose and control infection, including activities on the viral genome sequencing for epidemiological surveillance, COVID-19 patients with reinfections or vaccination failures and outbreak investigations.”
Line 56, COVID-19 should be first defined in line 31 where it is first used
AR
As suggested, we included the extension of the abbreviation COVID-19, that is Coronavirus disease 2019.
Line 64 “SARS-CoV-2 positive samples”
AR
We modified this expression, as required.
Line 69-70, would this study have benefitted from blinding, as tests were carried out after NGS? Please could the authors discuss or reply to this.
AR
Multiplex RT-PCR SARS-CoV-2 variants assays were performed after NGS without to blind the sequencing results with the aim to test a number and type of specimens reflecting those they were observed and sequenced during each Quick Survey.
Line 76, the authors should define “wild type”, in this case being B lineage from Wuhan?
AR
SARS-CoV-2 wild type group is characterized by all viruses which they are not viral variants and they were circulating in the same period of viral variants under analyses. These strains are missing of the Spike mutations of interest. It is composed by: B.1.177.15 - Clade 20E (N = 1), B.1.258 - Clade 20A (N = 2), B.1.160 - Clade 20A (N = 1), B.1.177 - Clade 20E (N = 6), B.1.177.75 - Clade 20E (N = 1). These data were included in the lines 76-79, as required.
Line 77 and 86, a small table of the used tests would be helpful
AR
As suggested, we included in this point of the text an additional table to detail the Spike protein mutations of each RUO assays, excluding data about the test that it is used for SARS-CoV-2 diagnosis. It is cited but it is not of interest for the study.
Line 86 onwards, a table highlighting the coverage of each RUO would be useful (including which mutations are detected, etc), this could be combined with the above comment
AR
A reported above, we have added a table to detail the Spike protein mutations of each RUO assays.
Line 210, a citation for WHO guidelines would be useful
AR
We modified this sentence.
Line 234, 31/72 is supposed to be 43% ?
AR
It was a calculation mistake. Now, we report the correct number of sex male cases (36 of 72, 50%).
Line 249-251, clarity needed here, was a patient tested daily for 59 days? And other patients too, with the majority testing positive by PCR for 17 +/- 8 days?
AR
In this lines we report that: "Mean duration of SARS-CoV-2 RNA shedding was 17 ± 8 days (maximum value 59 days). As expected, an increase was observed for patients group with severe symptom that it is 34 ± 17 days."
SARS-CoV-2 RNA shedding is the time between the first naso-pharyngeal (NF) swab positive and the first NF swab negative in addition to the asymptomatic status of the patient. Therefore, in this sentence we report how long viral load among our study cases. Patients group with severe symptom were positive for long times than to patients group that is asymptomatic.
Line 254 “achievement of vaccine efficacy” needs rewording, or clarification, do the authors mean the time from first (or full) vaccination to infection?
AR
We have clarified that vaccine efficacy is intended from one week after the second dose. Text was modified about it.
Line 258, why was the age of 50 used as a cut-off, are there references to cite for this decision?
AR
Age is by itself a risk factor for severity of and mortality due to COVID-19. Indeed, disease's severity and outcome vary with age and other factors. Subjects over 50 years of age with comorbidities are significantly more likely to be hospitalized for COVID-19 [1-2]
- Huang C, Wang Y, Li X, et al. Clinical features of patients infected with 2019 novel coronavirus in Wuhan, China. The Lancet. 2020;395(10223):497-506. doi:10.1016/S0140-6736(20)30183-5 2.
- Myers LC, Parodi SM, Escobar GJ, Liu VX. Characteristics of Hospitalized Adults With COVID-19 in an Integrated Health Care System in California. JAMA. 2020;323(21):2195-2198. doi:10.1001/jama.2020.7202
We have added these references in the text to support our decision to use 50 years of age as cut-off.
Line 267, mention of table 2, should this be before?
AR
In the result section, from line 285 we report for the first time diagnostic data of assays for screening of SARS-CoV-2 variants. Therefore, this table must be cited in this paragraph.
Line 274, two wrong mutations when compared to NGS data?
AR
We confirm that UltraGene Assay SARS-CoV-2 452R & 484K & 484Q Mutations V1 and SimplexaTM SARS-CoV-2 Variants Direct have detected two wrong mutations when compared to the NGS data.
Line 288, figure 1, please increase the text resolution of the mutations
AR
Font size of the Figure 1 was modified, as required.
Line 330, increase text resolution in fig 2 please
AR
Font size of the Figure 2 was modified, as required.
Line 330, it was previously reported that NGS occurred before RUO use, please clarify which way it is! The figure suggests RUO is carried out before NGS
AR
The figure 2 was elaborated as diagnostic algorithm to use on the basis of the data that we report with our experiences. Indeed, in the lines 328-330 we write it: “To optimize control of the SARS-CoV-2 variants, the molecular detection of the most widespread spike protein mutations could be used as test of second level in the diagnosic workflow of COVID-19 (Figure 2)”. The figure 2 was cited only in this sentence with the aim to propose the use of these test before the sequencing, and not after NGS as we performed in this study.
Line 345, the authors have not stated the costs of these tests, aside from terming them “cheap”,
AR
We define these assays only as less expensive without to report the exact cost because here, there are not commercial aims and furthermore, they have different costs. However, it is known that multiplex PCR, with single test cost about inferior to 50,00€, is cheap compared to the NGS, that it is paid 200,00€ - 300,00€ each sequence.
The lines cited in these responses are referred to the manuscript version without track changes.
